# Exploring the Potential of Lapatinib, Fulvestrant, and Paclitaxel Conjugated with Glycidylated PAMAM G4 Dendrimers for Cancer and Parasite Treatment

**DOI:** 10.3390/molecules28176334

**Published:** 2023-08-30

**Authors:** Łukasz Uram, Konrad Wróbel, Małgorzata Walczak, Żaneta Szymaszek, Magdalena Twardowska, Stanisław Wołowiec

**Affiliations:** 1Faculty of Chemistry, Rzeszów University of Technology, 6 Powstańcow Warszawy Ave., 35-959 Rzeszów, Poland; luram@prz.edu.pl (Ł.U.); mwalczak@prz.edu.pl (M.W.); 163835@stud.prz.edu.pl (Ż.S.); 164811@stud.prz.edu.pl (M.T.); 2Medical College, Rzeszów University, 1a Warzywna Street, 35-310 Rzeszów, Poland; konradwrobel300@gmail.com

**Keywords:** paclitaxel, fulvestrant, lapatinib, glycidylated PAMAM G4 dendrimer, U-118 MG glioma, A549 NSCLC, HaCaT keratinocytes, *Caenorhabditis elegans*, proliferation and viability assays, cellular uptake

## Abstract

Fulvestrant (F), lapatinib (L), and paclitaxel (P) are hydrophobic, anticancer drugs used in the treatment of estrogen receptor (ER) and epidermal growth factor receptor (EGFR)-positive breast cancer. In this study, glycidylated PAMAM G4 dendrimers, substituted with F, L, and/or P and targeting tumor cells, were synthesized and characterized, and their antitumor activity against glioma U-118 MG and non-small cell lung cancer A549 cells was tested comparatively with human non-tumorogenic keratinocytes (HaCaT). All cell lines were ER+ and EGFR+. In addition, the described drugs were tested in the context of antinematode therapy on *C. elegans*. The results show that the water-soluble conjugates of G4P, G4F, G4L, and G4PFL actively entered the tested cells via endocytosis due to the positive zeta potential (between 13.57–40.29 mV) and the nanoparticle diameter of 99–138 nm. The conjugates of G4P and G4PFL at nanomolar concentrations were the most active, and the least active conjugate was G4F. The tested conjugates inhibited the proliferation of HaCaT and A549 cells; in glioma cells, cytotoxicity was associated mainly with cell damage (mitochondria and membrane transport). The toxicity of the conjugates was proportional to the number of drug residues attached, with the exception of G4L; its action was two- and eight-fold stronger against glioma and keratinocytes, respectively, than the equivalent of lapatinib alone. Unfortunately, non-cancer HaCaT cells were the most sensitive to the tested constructs, which forced a change in the approach to the use of ER and EGFR receptors as a goal in cancer therapy. In vivo studies on *C. elegans* have shown that all compounds, most notably G4PFL, may be potentially useful in anthelmintic therapy.

## 1. Introduction

Today, cancer is a very serious problem in the world. According to statistics from the American Cancer Society (ACS), in 2023, lung and bronchus cancer will be the third most frequently diagnosed and the most deadly type of cancer in the USA [1]. Patients with stage I lung cancer have a five-year survival of 68.4%, whereas patients with stage IV lung cancer have a five-year survival of only 5.8%. Standard therapies for stages I—IIIA include surgical resection and then adjuvant systemic therapy, e.g., with cisplatin [2]. Despite the use of post-resection chemotherapy, about half of patients with stage IB cancer and roughly three-quarters of patients with stage IIIA lung cancer suffer from a metastatic disease [3]. Despite the lower incidence, glioblastoma multiforme is also characterized by high mortality. Glioblastoma multiforme (GBM) is a highly aggressive grade IV astrocytoma, according to the World Health Organization, representing 15 to 20% of all primary intracranial neoplasms in adults [4]. Statistically, patients usually die within 15–16 months of being diagnosed, and treatment with available methods (surgical resection, radiotherapy and chemotherapy) allows 5% of patients to survive for 5 years from diagnosis [5]. Currently, the only widely available chemotherapeutic agent is temozolomide, but its effectiveness is far from satisfactory, with numerous side effects [6]. Therefore, the development of new therapies and drugs against these cancers remains an extremely important issue.

Glioblastoma multiforme is approximately 1.6 times more common in males than females and is therefore suspected to be related to sex hormone levels, particularly 17 β-estradiol (E2) [7]. Moreover, women have a better response to treatment than men, indicating that estrogen could possibly have a protective function [8]. It has been shown that cell proliferation, migration, and invasion are promoted by E2 in human glioma cells, with a concentration-dependent effect [9,10]. Estrogen acts on cells via estrogen receptors (ERs). The type of ER is also important. ERα signaling has been found to be associated with glioma progression, but ERβ has anti-cancer effects [11]. Glia cells exhibit lower expression of ER⍺ than that of low-grade gliomas, but high-grade gliomas decrease such expression. On the contrary, the expression of ERβ is higher in glia cells than that in gliomas [8].

Some studies have proven that estrogen, interacting with its receptor, can stimulate the proliferation, apoptosis resistance, angiogenesis, migration and metastasis of lung cancer cells. ERs are widely expressed in the lung epithelium [12]. Estrogen causes the overexpression of pro-inflammatory proteins and ligands that promote tumor resistance. It is now recognized that non-small-cell lung cancer (NSCLC) is an ER-positive cancer. Estrogen receptor alpha (ERα) and beta (ERβ) have been identified in both NSCLC cell lines and tissues. ERβ is the predominant type of ER in NSCLC and is overexpressed in 60–80% of lung cancer tissues. The ERβ protein has been presented at higher levels in lung cancer cells than in normal lung tissue from the same patients [13]. Estrogen receptor β might also impact the progression and invasion of NSCLC, although the mechanism still requires elucidation [14]. Therefore, it seems right to investigate the effectiveness of inhibitors of ERs in glioma and non-small-cell lung cancer therapy. One of the candidates for a potential anticancer agent is fulvestrant, which is commonly used in the treatment of metastatic hormone receptor-positive breast cancer. After binding to an ER, it blocks and degrades it and, as a consequence, inhibits estrogen signaling through the ER [15]. About 80% of breast cancers express ERs, with their survival and proliferation driven by estrogen acting as a ligand and binding to the ER, which is then translocated to the cancer cell’s nucleus [16]. Unlike other selective estrogen receptor degraders (such as tamoxifen), fulvestrant almost completely inhibits estrogen signaling through the ER [17]. Hamilton et al. proved that fulvestrant reduces the mesenchymal features of lung cancer cells and sensitizes the cells to chemotherapy [18].

Another factor that is specific to many types of cancer, including GBM and NSCLC, is the epidermal growth factor receptor (EGFR). The EGFR is a member of the ERBB family of tyrosine kinase receptors (EGFR/ErbB-1, HER2/ErbB-2, HER3/ErbB-3, and HER4/ErbB-4) and is a trans-membrane glycoprotein, consisting of an extracellular ligand-binding domain and a cytoplasmatic tyrosine kinase domain. The EGFR signaling cascade is a key regulating factor in cell proliferation, apoptosis, metastasis and angiogenesis, differentiation, division, survival, and cancer development [19,20]. The over-activation of EGFR is observed in many types of cancers, including head, neck, breast, lung, colorectal, prostate, kidney, pancreas, ovary, and brain cancers [21]. This phenomenon was also described in GBM and NSCLC [22,23]. The therapy of tumors with EGFR overexpression is disappointing; therefore, anti-EGFR agents are recommended in the first-line treatment [20] because EGFR activity may be controlled by binding to the tyrosine kinase domain [24]. EGFR is most amplified in GBM, among all receptor tyrosine kinases (57% of patients) [25]. This leads to overexpression of EGFR protein and increased tumorigenesis and progression of glioma [26]. However, in some cases, overexpression of EGFR was present without EGFR gene amplification [27].

Lapatinib is a reversible, first-generation inhibitor of EGFR type 1 (HER1) and type 2 (HER2), administered orally and approved for the treatment of breast cancer overexpressing HER2 in combination with capecitabine [28]. Some studies indicated that lapatinib enhances 5-aminolevulinic acid-mediated protoporphyrin IX fluorescence and photodynamic therapy response in human glioma cell lines [29]. Moreover, the addition of pulse high-dose lapatinib to standard therapy for newly diagnosed GBM was largely well tolerated and safe [30]. Satisfactory results in glioma therapy were also observed after the conjugation of lapatinib with nanoparticles. Lapatinib-incorporated lipoprotein-like nanoparticles significantly inhibit the growth of glioma U87 xenografts and were effectively taken up by glioma cells. The required cumulative dose of LTNPs was only 5% compared to lapatinib alone [31]. Unfortunately, there is a report that lapatinib did not show significant activity in GBM patients in phase II/III clinical trials [32]. Human HER2 has emerged also as a therapeutic target of interest for non-small-cell lung cancer in recent years. HER2 alterations are present in 7–27% of de novo NSCLC and may serve as a resistance mechanism in up to 10% of EGFR-mutated NSCLC [33]. However, lapatinib as a single agent or combined with pemetrexed did not result in a sufficient antitumor response in NSCLC [34], but it is worth mentioning that lapatinib had a similar response rate to pemetrexed monotherapy [35].

Recent studies have shown an interaction between the EGFR pathway and ER signaling in lung adenocarcinoma; therefore, combined therapy with tyrosine kinase inhibitors and antiestrogen therapy is a promising solution. A combination of the ER antagonist and the EGFR tyrosine kinase inhibitor caused a decrease in cell proliferation and lung tumor growth in both in vitro and in vivo studies [36,37,38,39]. Thus, it is speculated that a similar effect may be achieved with combined therapy against glioblastoma multiforme since it is known that cells of this malignant tumor manifest changes in the ER and EGFR.

Another drug used in the treatment of NSCLC is paclitaxel, which facilitates microtubule polymerization and disturbs cellular functions, especially normal cell division, and causes cell apoptosis [40]. Paclitaxel is administered alone or with platinum compound in advanced NSCLC [41,42]. Moreover, this drug has clinically relevant activity in heavily pretreated small-cell lung cancer [43]. Low-dose weekly paclitaxel was active and well tolerated as second-line chemotherapy in a non-selected patient population with advanced NSCLC [44]. Furthermore, it is known, that paclitaxel shows anticancer effects against NSCLC by activating apoptosis through enhancing ROS generation, inducing cell cycle arrest, and suppressing EGFR/PI3K/AKT/mTOR signaling pathway [45]. Paclitaxel is also a potential candidate for anti-glioma therapy with 1400-fold lower IC_50_ than the most used temozolomide in in vitro studies. Several clinical trials evidenced that paclitaxel caused minimal responses in glioma treatment, due to the presence of a protective structure in the brain—the blood–brain barrier (BBB) [46,47,48,49,50]. Moreover, the multidrug resistance limits the clinical applications of paclitaxel drug, due to the presence of P-glycoprotein in NSCLC cell membrane and BBB [40]. One possible approach is to use a vehicle like PAMAM dendrimers that deliver paclitaxel, fulvestrant, and lapatinib across BBB. PAMAM dendrimers are symmetric, hyperbranched, polymeric macromolecules with polyvalent cores and dendritic branches, with well-defined nano-sized globular structures. PAMAMs are promising vehicles for efficient anticancer drug delivery. Due to the high-density surface functionalities, PAMAMs allow for the attachment of suitable ligands for cancer targeting, imaging, and therapy. Moreover, they enhance the solubility and bioavailability of carried drugs. The nano-size and ability of PAMAMs to interact with plasma proteins drive the passive targeting through the enhanced permeability and retention effect (see review [51]). In the case of brain tumors or metastases, anticancer therapeutics often fail, mainly due to the inability to cross the BBB. As a result, cancer cell drug uptake and drug resistance occur [52]. In order to eliminate the toxicity of PAMAM dendrimers and increase their permeability, their amino surface groups are hydroxylated with PEG or glycidol [53,54].

In this study, we used non-toxic, glycidylated fourth-generation (G4) PAMAM dendrimers as the carrier for paclitaxel (P), fulvestrant (F), and/or lapatinib (L). We have synthesized four PAMAM conjugates, G4P, G4F, G4L, and G4PFL, and studied their efficiency in anticancer therapy against ER-positive and EGFR-positive A549 non-small-cell lung cancer and U-118 MG glioma cells, and comparatively with ER-positive and EGFR-positive immortalized, non-cancerous human keratinocytes (HaCaT). We hypothesized that the G4PFL conjugate components would have an additive or even synergistic effect.

Potential anticancer therapeutics require also characterization of toxicity and mutagenicity in vivo. The use of vertebrate-based models should be preceded by tests on invertebrate animals. The nematode *Caenorhabditis elegans* is a very attractive animal model to characterize the properties of anticancer therapeutics. The small size, ease of handling, and possibility of genetic modification of *C. elegans* provide a sophisticated in vivo platform that combines the technical advantages of a microorganism with the greater biological complexity of a multicellular organism [55,56]. *C. elegans* has been used successfully to test the activity of EGFR tyrosine kinase inhibitors [57], selective estrogen receptor degraders (SERDs) [58], and microtubule-stabilizing drugs [59]; therefore, the biological activity of the synthesized constructs was studied also in vivo on *C. elegans*. Additionally, its usefulness as a model for mixture toxicity testing was confirmed [60].

## 2. Results and Discussion

### 2.1. Syntheses and Characterization of Dendrimer Conjugates

The conjugates of three anticancer drugs, paclitaxel (P), fulvestrant (F), and lapatinib (L), were obtained by attachment of functionalized drugs by amide bonds into primary amine groups of PAMAM G4 (Figure 1 and Appendix A).

Thus, P was converted into 2′-O-succinate (P-suc) as described in [61] and the latter was activated with 1-methyl-2-chloropyridin iodide and linked into PAMAM G4 by an amide bond to get G4 substituted with average four P-suc residues (G4^4P^) as described before [61] (Appendix A). The conjugate was labeled with one FITC and the remaining primary amine groups were converted by the addition of *R*-glycidol to obtain G4*^4P109gl^. This conjugate was characterized by the ^1^H NMR spectroscopy (Figure 2A).

The conjugates of L and F with PAMAM G4 were obtained from L-N(7)-(p-nitrophenylcarbonate) and F-O(17)-(p-nitrophenylcarbonate), which were adequate to react quantitatively with primary amine groups of single FITC-labeled PAMAM G4 (G4*), as described previously for PAMAM G3 [62] (Appendix A). Thus, high-substituted conjugates were obtained, which were further converted by *R*-glycidylation into G4*^13L85gl^ and G4*^13F64gl^ derivatives. The stoichiometry of conjugates was determined by the ^1^H NMR spectroscopy (Figure 2B,C, respectively) by comparing the intensity of the PAMAM G4 resonance at 2.2 ppm (corresponding to [248H] internal reference) and resonances of 3′^P^-H ([4H]) in the case of G4*^4P109gl^, 3^L^-H ([13H]) in the case of G4*^13L85gl^, and 1^F^-H ([13H]) in the case of G4*^13F64gl^. The number of 2,3-dihydroxypropyl substituents (gl) was determined by the integration of 2^gl^ and 3^gl^ resonances in the 3.3–3.6 ppm region.

The ternary conjugate, containing 4P, 11L, and 11F residues was obtained from G4*^4P^ conjugate by reaction with NPCF-activated equimolar mixture of L and F followed by *R*-glycidylation. The stoichiometry of the product, G4*^4P11F11P74gl^ was determined by the ^1^H NMR spectroscopy, while spectral assignment was performed with corresponding ^1^H-^13^C heteronuclear HMQC and HMBC NMR experiments (Figure 2D and Appendix A).

For further discussion the abbreviations for primary, secondary, and tertiary conjugates were simplified as follows: G4P–(G4*^4P109gl^), G4F–(G4*^13L85gl^), G4L–(G4*^13L85gl^), G4PF–(G4*^4P10F98gl^), G4PL–(G4*^4P10L98gl^), and G4PFL–(G4*^4P11F11P74gl^) as described in the first column of Table 1.

The primary, secondary, and tertiary conjugates were well soluble in DMSO-d_6_ (up to 5 mM concentration). The hydrodynamic diameter of obtained materials was determined for 1 µM aqueous solutions in water and in phosphate buffer pH 5. The diameter of single-FITC labeled perglycidylated carrier (G4*^126gl^, Table 1) without attached hydrophobic drug molecules was found ca 5 nm in water, which is ca 0.5 nm larger than for commercially available PAMAM G4. We found that all conjugates were associated with water (Table 1).

The size of nanoparticles was at least one order of magnitude larger than expected for mono-molecular dispersion (about 5 nm diameter). The number-averaged diameters of conjugates (d(N)) are listed in Table 1. The volume-averaged diameters were significantly larger due to relatively high polydispersion (PDI > 0.1) (Appendix A). Furthermore, the size of nanoparticles in phosphate-buffered dispersion pH 5 was different, presumably due to the protonation of PAMAM G4 internal tertiary amine groups. Eventually, all associates of conjugates indicated highly positive zeta potential at pH 7 (Table 1 and Appendix A), which grew even higher at pH 5 (Appendix A). The change of nanoparticle size in pH 5 was observed particularly in the case of G4L and G4F, which showed PDI > 0.23 and complicated pattern of d(V) ranging from 40 to 1000 nm particles (Appendix A). Generally, d(V) diameters were considerably higher than d(N) diameters for all conjugates (Appendix A). Another parameter showed regular behavior, namely zeta potential, which was considerably higher in acidic (pH 5) conditions than in neutral water (Appendix A). This was attributed not only to the protonation of macromolecules in nanoparticles, but also to different nanoparticle sizes. The size of associates was stable (Appendix A) in water. However, in the case of G4F further association of macromolecules upon pH decrease from 7 to 5 was striking (Appendix A).

### 2.2. Cellular Accumulation

Three cell lines were selected as a therapeutic target for the glycidylated PAMAM G4 conjugates with paclitaxel, fulvestrant, and/or lapatinib: A549 non-small-cell lung cancer, U-118 MG glioma cells, and comparatively non-cancerous, immortalized human keratinocytes. It was reported that A549 and HaCaT cells express estrogen receptors: ERα and ERβ [63,64]. EGFR is overexpressed in A549 cells [65] and expressed in HaCaTs [66]. Although there is a lack of information concerning ER expression in U-118 MG glioma cells, in cytogenetically similar U-138 MG glioma, ER expression was identified [67]. U-118 MG glioma cells express EGFR (about 85% of cells) and HER2 (about 20% of cells) [68]. Therefore, these cell lines were proper models of cancerous or non-cancerous cells for testing.

One of the most important properties of a drug carrier is its ability to effectively penetrate cells. As we showed earlier, PAMAM dendrimers flanked with *R*-glycidol residues were non-toxic up to 300 µM concentration and efficiently penetrating carriers, with selectivity to cancer cells (SCC-15) against HaCaTs [53]. Physicochemical properties, such as particle size, shape, and surface charge, play a key role in the cellular uptake of nanoparticles [69]. Fluorescently labeled G4P, G4F, G4L, and G4PFL were taken up and accumulated inside all tested cell lines in the cytoplasmatic area, but not in nuclei already at low, nanomolar concentrations (Figure 3 and Figure 4).

An important factor affecting the ability of nanoparticles to penetrate cells is the value of their zeta potential [69]. Nanoparticles should have a positive zeta potential to penetrate the negatively charged cells efficiently [70]. DLS measurements showed that the tested nanoparticles had positive zeta potentials in the range of 13.57–40.29, which is proper for active penetration into the cells. It is worth mentioning that the glycidylated PAMAM G4 dendrimer average diameter is equal to 6.21 nm. All studied conjugates had a diameter in the range of 99.20–138.23 nm; therefore, we concluded that they were associated with water, which was probably a factor limiting their uptake.

Among all cell lines, HaCaT cells showed the highest uptake and accumulation of all conjugates (Figure 3). A slightly weaker degree of absorption was observed in cells from lines A549 (G4P and G4PFL) and U-118 MG (G4F and G4L). Due to the use of different concentrations of drugs, it is difficult to compare the absorption of individual conjugates. However, if we focus on the analysis of the fluorescence intensity of conjugates in the range of 320–625 nM concentrations (about 400 nM) (Figure 3), we can notice some patterns. Keratinocyte cells accumulated similar amounts of all conjugates used. G4F conjugate was stronger absorbed in glioblastoma than G4P and G4PFL, which may be related to drug resistance and efflux, due to the presence of paclitaxel, since glioma cells express P-glycoprotein responsible for this phenomenon [71,72]. Similarly, non-small-cell lung cancer absorbed mainly G4F and least G4P.

In this study, we have used a glycidylated dendrimer, which was supposed to reduce the uptake of conjugates by HaCaT cells, compared to the tumor cells [53]. Differences in the degree of absorption of individual conjugates are difficult to explain, as they may be related to ER-mediated endocytosis [73] or EGFR-mediated endocytosis (mainly clathrin-mediated endocytosis) [74,75] and the processes of regurgitation by P-glycoprotein, which are present in the tested cell lines [72,76]. The possibility of penetration of the studied nanoparticles by clathrin-mediated endocytosis is confirmed by the fact that it concerns nanoparticles with a diameter of approx. 100–150 nm. This was the diameter of the aggregates of all studied conjugates (Table 1). Additionally, fast endophilin-mediated endocytosis (FEME) was recently discovered, which is a non-constitutive process that is triggered upon activation of specific receptors including EGFR, and the conjugate of paclitaxel to human albumin is internalized by the cells through caveolae-mediated endocytosis [77].

### 2.3. Cytotoxicity

Cytotoxicity of studied compounds was estimated with two different tests—neutral red assay (NR) and tetrazolium salt assay (XTT). The more sensitive NR assay indicated that among drugs alone the most effective in cell killing was paclitaxel with the IC_50_ equal to 50.35, 55.56, and 2.53 nM for A549, U118 MG, and HaCaT cells, respectively (Figure 5). 

Lapatinib showed a weaker effect with meaningly higher values of IC_50_ coefficients equal to 1.67, 14.9, and 6.2 µM (Table 2). Fulvestrant was not toxic to any of the tested cell lines up to 100 micromolar concentration, which was described by others for A549 cells [78].

Obtained results of toxicity are rather consistent with those reported by others. IC_50_ of paclitaxel for A549 and U-118 MG cells after 48 h incubation was equal to 11.0 and 21.1 nM, respectively [80,81]. IC_50_ of fulvestrant for A549 after 48 h incubation was over 10,000 nM [78]. In the case of lapatinib, IC_50_ was in the range of 5000–10,000 nM for A549 (48 h incubation), 8000 nM for U-118 MG, and 200 nM for HaCaT (72 h incubation) [82,83,84].

The degree of absorption of the tested conjugates was reflected in the toxicity induced by them in particular cell types. The most sensitive HaCaT cells accumulated the highest level of all dendrimer conjugates. Glioma cells took mainly G4L and G4F conjugates, which caused a stronger effect than in lung cancer. A549 cells were more destroyed than glioma cells due to the highest uptake of G4P and G4PFL conjugates. One of the desired effects of dendrimers is to enhance the action of drugs attached to them. In our study, the binding of 13 residues of F to the glycidylated PAMAM G4 dendrimer resulted in an approximately 2-fold decrease in IC_50_ values for HaCaT dermal keratinocytes (comparing the concentration of introduced drugs). Such an effect was not observed in the A549 cell line, where a decrease in cell viability under 50% was not noticed. The paclitaxel-containing conjugate did not induce a decrease in cell viability with increasing concentrations in HaCaT and A549 cells. This phenomenon was described previously as characteristic of paclitaxel [85].

In the case of lapatinib conjugate, its action was 2- and 8-fold stronger against glioma and keratinocyte cells, respectively, than the equivalent of lapatinib alone (Figure 5 and Table 2). Substitution of modified G4 PAMAM dendrimer vehicle with 4 residues of P resulted in about 2-fold lower attenuation of the conjugate effect compared to the drug alone for HaCaT and U-118 MG cell lines, and even 25 fold for A549 cells (Figure 4). In the remaining cases, the effect of conjugates was proportional to the concentration of the introduced drugs.

We also assumed that G4PFL conjugate containing all three tested drugs would enhance the therapeutic effect. The G4PFL compound was more efficient in killing the tested cells than the G4F and G4L conjugates, but less than G4P. It can therefore be presumed that its effectiveness came mainly from the attached paclitaxel residues.

The lower activity of G4PFL than G4P may be caused by the differences in diameter and physico-chemical properties of both conjugates. G4PFL contains as many as 26 molecules of drugs with highly hydrophobic properties on the surface, which significantly cover glycidol residues, which are responsible for the effect of selectivity to cancer cells, and have a high nanoparticle diameter of 105.47 nm. The nanoparticles of G4P are smaller (99.20 nm in diameter) and can penetrate more easily than those of G4PFL. The G4F with the largest diameter (138.23 nm) penetrated weakest. However, the G4L conjugate with a diameter of about 113.13 nm was not more potent than G4PFL. It is therefore necessary to conclude that the activity of the tested conjugates is related to the combined effect of their nanoparticle size and the activity of the drugs attached to them.

The more sensitive NR assay measures lysosomal integrity, which is the indicator of cell health [86]. The second assay used, the reduction of tetrazolium salts (XTT), evaluates the reducing properties of trans-plasma membrane electron transport including the activities of mitochondrial oxidoreductases; therefore, it is a good indicator of the mitochondria condition [87]. The results obtained with the XTT assay revealed a similar toxicity pattern as the NR assay (Figure 6). 

Comparing the IC_50_ values, it can be observed that the conjugate with lapatinib (G4L) more strongly disturbed the condition of lysosomes and cell membranes, while the conjugates with paclitaxel and fulvestrant affected the dysfunction of mitochondria mainly by changing the activity of mitochondrial oxidoreductases (Table 2). 

### 2.4. Proliferation

Microscopic observations during NR assay indicated that studied conjugates not only destroyed and killed cells, but also inhibited their proliferation. In order to confirm the anti-proliferative effect of the conjugates used, a proliferation test consisting of measuring the amount of DNA, which is proportional to cell number, was performed [88]. Paclitaxel, fulvestrant, and lapatinib are well-established anti-proliferative agents [45,89,90]. All studied cell lines were incubated for 72 h with G4P, G4F, G4L, and G4PFL solutions to test their anti-proliferative effects, since doubling time of A549, U-118 MG, and HaCaT was equal 20, 35, and 24 h, respectively.

The greatest reduction in proliferation was observed in HaCaT cells, where G4P was most active, causing an 80% decrease in cell number from 1.25 nM concentration compared to the untreated control. The G4PFL and G4L performed lower activity, and G4F was the weakest one (Figure 7). 

Lung adenocarcinoma A549 cells were significantly less inhibited, with G4P showing the highest activity, followed by G4PFL, G4L, and G4F, respectively. The G4L conjugate even increased the proliferation up to 20% at 100 and 400 nM concentrations. The phenomenon of proliferation promotion induced by lapatinib was described before for breast cancer cells [91]. The U-118 MG glioma cells did not respond to either G4P or G4F in the tested concentration range. G4PFL inhibited slight proliferation only at 25 nM concentration by 20%. G4L stopped cell growth to nearly zero already from a high 1600 nM concentration, and a similar pattern of activity to that of A549 cells was seen (Figure 7).

In conclusion, the toxicity profile of all tested conjugates against HaCaT and A549 cells was connected rather with the anti-proliferative action of the studied compound. Meanwhile, in glioblastoma, cytotoxicity was associated with cell damage via impaired mitochondrial activity and ATP levels as demonstrated in the NR and XTT assays. Only the cytotoxic effect of G4L was anti-proliferative (compare Figure 5, Figure 6 and Figure 7). 

### 2.5. Effect on the Worm Survival

Today, anthelmintic drugs are considered to be potentially repurposed anticancer drugs. For example, it was indicated that flubendazole inhibits tubule polymerization and angiogenesis stimulate apoptosis, ferroptosis, autophagy, and cancer stem-like cell killing and tumor degeneration [92]. The opposite effect is also possible—anticancer drugs (especially those without too strong side effects) may become anthelmintic agents in lower doses. *Caenorhabditis elegans* responds to estrogenic hormones and possess estrogen receptor [93,94]. Also, EGFR receptors were found in *C. elegans* [95,96]. Therefore, it seems that PAMAM G4 conjugates with estrogen receptor degrader (fulvestrant) and epidermal growth factor receptor inhibitor (lapatinib) may potentially be a way to create the anthelmintic drugs.

*Caenorhabditis elegans* viability was assessed after seven days of incubation with G4P, G4F, G4L, and G4PFL. Results indicated that all used conjugates were highly toxic at studied concentrations. The lower toxic effect was observed after G4P treatment with LC_50_ over 24 μM concentration (Table 3). 

Seven days of incubation caused a statistical decrease of nematode viability of over 40% at 6 and 12 μM concentrations. The number of viable individuals did not differ drastically between used concentrations of G4P, with the exception of 12 and 6 μM concentrations, where sharp decreases in viability occurring on the second and third day of incubation were observed, respectively (Figure 8).

The G4F and G4L decreased the *C. elegans* viability stronger, with LC_50_ equal to 12.50 and 14.80 μM, respectively. The most toxic concentration of fulvestrant was 15 μM during six days of observation, with the highest mortality at 30 μM concentration of G4F on the last day of the experiment. G4L induced lowering of nematode viability with increasing drug concentrations. The highest drug concentration (30 μM) caused a decrease in viability to 37.5% after 7 days of incubation. The G4PFL was the most toxic conjugate with 6.75 μM LC_50_ after seven days of incubation, and its toxicity was only slightly weaker than a common anthelmintic drug, mebendazole (LC_50_ = 4.0 μM). The highest, 10 μM concentration of conjugate containing all three used drugs caused a lowering of the *C. elegans* viability to below 20% (Figure 8). Thus, an additive effect of the drugs attached to the G4PFL multi-component conjugate can be seen.

Morphological and behavioral changes during observation were also noticed in the case of all used conjugates, which were the most visible in the group treated with G4P, and the least for G4F. Most often, the degradation and deformation of internal organs, as well as a reduction of nematode motility sometimes similar to convulsions were observed. In the G4P group, degradation occurred mainly on the side of the head and the nerve ring. In G4F degradation, it was more frequent from the tail side. G4L (at 7.5 μM and 30 μM concentrations) deformed cuticle and internal organs. These changes are visible for the most part throughout the body of the nematodes. In addition, the nematode body seemed to shrink inside the cuticle, and a molting process was also observed (Figure 9). 

It can also be assumed that shrinking was part of molting. This was proven by observing the same individual for two consecutive days. On the first day, shrinkage was visible, but on the next day, there was no cuticle, which had already separated from the nematode’s body. This process should not take place at this stage of a nematode’s life and probably was caused by conjugates since adult nematodes do not molt [97]. 

All studied conjugates caused a significant reduction of nematode length for about 1.3–1.7 fold. The body size of nematodes depends on genetic and environmental factors [98]. Post-larval C. elegans size is related to the process of endoreduplication that occurs in the hypodermis [99]. Mature C. elegans continues to grow in the absence of cell divisions [100]. It can be concluded that the tested drugs impact the pathway associated with cell growth, by degrading the epidermis and affecting the hypodermis. Nematode cuticles mostly consist of collagen (more than 80%) [101]. It has been shown that epidermal collagen genes can act as positive regulators, dose-dependent regulators, and negative regulators of body size [102]. In connection with the damage to the cuticle, which was observed, disruption of epidermal gene expression may caused inhibition of C. elegans growth.

**Conclusions:** Obtained conjugates efficiently penetrated all tested cells at relatively low concentrations, despite the fact that they formed associates. Due to their properties and presence of glycidol, they have the potential to become drugs in the treatment of brain tumors, not only such as glioblastoma, but also lung cancer, which has the ability to metastasize to the area of the central nervous system.

The activity of the tested G4P, G4L, and G4PFL conjugates was high already at nanomolar concentrations, but targeting them to the tumors expressing ER or EGFR destroyed not only tumor cells, but also non-tumorogenic ones, showing the presence of these receptors. Therefore, not only the desired effects, but also the side effects of such therapy should be considered.

The most promising seems to be the G4L conjugate, which showed 2- and 8-fold higher toxicity against glioma and HaCaT cells, respectively, than the equivalent of lapatinib alone. The tested conjugates (especially G4PFL) could be also considered for anthelmintic therapy since they caused significant damage and mortality to *C. elegans*—model nematodes for these diseases.

## 3. Materials and Methods

### 3.1. Dendrimer Synthesis and Characterization

All the chemicals used during the synthesis of PAMAM G4 dendrimer and its conjugates were purchased from Merck KGaA (Darmstadt, Germany). PAMAM G4 dendrimer was obtained at a 2 millimolar scale according to the procedure published by Tomalia et al. [103] and stored as a 15 mM solution in methanol to obtain conjugates with fulvestrant, paclitaxel, and lapatinib (AmBeed, Arlington Hts, IL 60004, USA). Further, 4-nitrophenyl chloroformate (NPCF), 2-fluorescein isothiocyanate (FITC), 4-dimethylaminopyridin (DMAP), and 1-methyl-2-chloropyridinium iodide (pyI) were purchased from Merck KGaA (Darmstadt, Germany). Spectra/Por^®^ 3 RC dialysis membrane (cellulose, MW_cutoff_—3.5 kDa) was provided by Carl Roth GmbH & Co., KG (Karlsruhe, Germany). 

#### 3.1.1. Spectroscopy

The ^1^H, ^13^C NMR 1D, and 2D spectra (^1^H-^1^H COSY and ^1^H-^13^C HSQC and HMBC) were recorded with Bruker 300 MHz instrument (Rheinstetten, Germany). 

#### 3.1.2. Conjugate Size and ζ Potential

Dynamic light scattering (DLS) and ζ potential of PAMAM conjugates were measured at pH 5 (in 0.05 M acetate buffer) and in water using Zetasizer Nano ZS instrument (Malvern, UK) at the concentration of 1 mg/mL (ca 0.05 mM solutions), as previously described [62].

### 3.2. Syntheses of Conjugates

#### 3.2.1. Conjugates of Paclitaxel (P), Lapatinib (L), and Fulvestrant (F) (Primary Conjugates)

In order to covalently attach P, L, and F the drugs were converted as follows: (a) 68.3 mg P (80 µmoles) was dissolved in 2 mL DMSO followed by the addition of 9.6 mg succinic anhydride (96 µmoles) in 100 µL pyridine. The mixture was heated at 60 °C for 12 h. Then, to this solution, 29.3 mg DMAP (240 µmoles) and 30.6 mg 2-chloro-1-methylpyridynium iodide (120 µmoles) were added while vigorously stirring and left for 4 h in darkness.

Then, activated paclitaxel succinate was added dropwise into 284.3 mg PAMAM G4 in 1 mL methanol. The mixture was left overnight at room temperature, transferred into a dialytic tube (cellulose; MW_cutoff_ = 3.5 kDa), and dialyzed for 3 days against water. The solution from the dialytic bag was transferred into the round bottom flask, water and other volatiles were removed on a rotary evaporator (pressure 10 mbar), and the solid residue was then dried for 12 h under 0.1 mbar pressure and characterized by the ^1^H NMR spectroscopy. The macromolecular product was identified as G4 substituted with four P equivalents attached by an amide bond through a succinate linker, G4^4P^. This product was further labeled with one equivalent of FITC (attached through thiourea bond) in methanol (5 mL) and then the remaining amine groups were reacted with *R*-glycidol (200 µL) at ambient temperature for 36 h. The final product was dialyzed and dried as before. Based on the ^1^H NMR spectroscopy, the primary paclitaxel conjugate average stoichiometry corresponded to 4 succinate paclitaxel, 1 fluorescein, and 104 2,3-dihydroxypropyl residues, G4*^4P104gl^. The theoretical molecular weight for the G4*^4P104gl^ was 17.47 kDa. Yield: 151.8 mg (8.68 µmoles); 43.4% calculated for starting G4.

PAMAM G4 dendrimer was single labeled with FITC (G4*) at 100 µmolar scale and stored in methanol solution for further syntheses of primary and ternary conjugates.

Lapatinib and fulvestrant were both functionalized by condensation with acid chloride, NPCF, as described elsewhere [56]. Briefly, 116.2 mg L (200 µmoles) was dissolved in 4 mL CHCl_3_ and 50 µL pyridine. After dissolution of L 45.0 mg NPCF was added and the mixture was refluxed for 1 hr. Then, volatiles were removed under reduced pressure and the solid residue was dissolved in 1 mL DMSO. The N-7 functionalized L was added dropwise into 141.9 mg G4* (10 µmoles) in methanol (1 mL). The reaction mixture was left at 45 °C in darkness for 12 h and the product was worked up by dialysis followed by evaporation of volatiles. Then, the conjugate was dissolved in methanol (255.7 mg in 5 mL), 170 µL of R-glycidol was added and the mixture was left at room temperature for 30 h. After dialytic purification and drying 247.8 mg of solid product was collected and identified as G4*^13L60gl^ by the ^1^H NMR spectroscopy. The theoretical molecular weight for the G4*^13L85gl^ is 25.26 kDa. Yield: 247.8 mg (9.8 µmoles); 98% calculated per starting dendrimer.

Fulvestrant (121.4 mg; 200 µmoles) was dissolved in 2 mL CHCl_3_ and 50 µL pyridine; 44.3 mg NPCF (220 µmoles) was added into the solution and refluxed for 1.5 h. Afterward, volatiles were removed and the solid residue dissolved in 1 mL DMSO. This solution was added dropwise into the solution of 252 mg G4* (17 µmoles) and the mixture was left at 45 °C overnight. The conjugate was purified by dialysis, then dried and dissolved in methanol. To this solution, 170 µL of *R*-glycidol was added and the mixture was left at room temperature for 30 h. Then, the mixture was dialyzed against water as previously, the solid residue was dried under reduced pressure and the product was characterized by the ^1^H NMR spectroscopy as G4*^13F64gl^ (theoretical MW = 30.0 kDa). Yield: 279.2 mg 9.32 µmoles; 54.8%.

#### 3.2.2. Binary and Ternary Conjugates of G4* with Paclitaxel, Lapatinib and Fulvestrant

The G4*^4P^ was a substrate for further syntheses to obtain binary and ternary conjugates with L and F. Other substrates, namely L and F (200 µmoles both; 116.2 mg and 121.3 mg, respectively) were activated with NPCF (20% molarexcess) as described above and dissolved in DMSO (20 mM solutions). The binary conjugates were obtained by addition of 2.1 mL solution of activated F (or L) into 70 mg of G4*^4P^ (3.8 µmoles; MW = 18.3 kDa) in 1 mL DMSO, while the ternary conjugate was obtained by addition of 2.1 mL of both F and L into 70 mg of G4*^4P^ in 1 mL DMSO. Three separate mixtures reacted for 24 h at 45 °C, then transferred into dialytic tubes, purified as before, and then dissolved in methanol. Next, 200 µL of R-glycidol was added to these semi-products, and the mixtures were left for 2 days at room temperature. Further solvents and excess of *R*-glycidol were evaporated, and solid products dissolved in methanol and dialyzed against water as before. The solid products were isolated and characterized by the ^1^H NMR spectroscopy. Thus, the following binary and ternary conjugates were identified as G4*^4P10L98gl^, G4*^4P10F98gl^, and G4*^4P11L11F74gl^. Theoretical molecular weights were 31 713 Da, 31 971 Da, and 37 529 Da, respectively. The products were isolated with the following yields: 88.9 mg G4*^4P10L98gl^ (2.8 µmoles; 73.7%), 95.7 mg G4*^4P10F98gl^ (3.0 µmoles; 78.9%), and G4*^4P11L11F74gl^ (3.1 µmoles; 81.6%).

### 3.3. Biological Studies

#### 3.3.1. Biochemical Reagents, Cell Lines and Materials

Human cell lines: lung carcinoma epithelial cells (A549) and glioblastoma (U-118 MG) were purchased from the American Type Culture Collection (ATCC, Manassas, VA, USA). Human immortalized keratinocytes (HaCaT) were provided by Cell Lines Service (Eppelheim, Germany). Dulbecco’s Modified Eagle’s Media (DMEM) and fetal bovine serum (FBS) were purchased from Corning Inc. (New York, NY, USA). Penicillin and streptomycin solution, phosphate-buffered saline (PBS) with and without magnesium and calcium ions, and Hoechst 33342 were provided by Thermo Fisher Scientific Inc. (Waltham, MA, USA). Trypsin–EDTA solution, hydrocortisone and 0.33% neutral red solution, phenazinemethosulfate (PMS), 0.4% trypan blue solution, dimethylsulfoxide (DMSO) for molecular biology, 5-Fluoro-2′-deoxy-uridine (FUdR), and other chemicals and buffers were purchased from Merck KGaA (Darmstadt, Germany). The XTT sodium salt was provided by Cayman (Ann Arbor, MI, USA). Cell culture dishes and materials were from Nunc (Roskilde, Denmark) or Corning Inc. (New York, NY, USA). Reagents used to culture *C. elegans* nematode were supplied by Sigma-Aldrich (Saint Louis, MO, USA) or Carl Roth GmbH & Co., KG (Karlsruhe, Germany).

#### 3.3.2. Cell Cultures

Three human cell lines: A549 (non-small-cell lung cancer cells), U-118 MG (glioblastoma multiforme, grade IV), and HaCaT (immortalized keratinocytes) were grown in DMEM supplemented with heat-inactivated 10% FBS and 100 U/mL penicillin and 1% streptomycin solution. Cells were cultured at 37 °C in a humidified 95% air, 5% CO_2_ with growth media changed every 2–3 days. Cells were passaged at 70–85% confluence with 0.25% trypsin–0.03% EDTA in PBS without calcium and magnesium ions. The morphology of cells was checked under a Nikon TE2000S Inverted Microscope with phase contrast (Tokyo, Japan). The number and viability of cells were estimated by a trypan blue exclusion test using an Automatic Cell Counter TC20 (BioRad Laboratories, Hercules, CA, USA). The working solutions of the synthesized dendrimer conjugates and the drugs alone were prepared in cell culture media with an adjusted concentration of DMSO (not higher than 0.1%). Control samples with non-treated cells in a complete culture medium with adjusted DMSO concentration were included in all biological assays.

#### 3.3.3. Toxicity Assays

Neutral red (NR) and XTT assay, A549, U-118 Mg, and HaCaT cells, were seeded in flat-bottom 96-well culture plates at a density of 1 × 10^4^ cells/well (100 µL cell suspension per well) and allowed to attach for 24 h. After cell culture removal, the studied conjugates of dendrimers or drugs alone were added in the range of increasing concentrations (100 µL cell per well) and cells were treated for 48 h at 37 °C. Then, the neutral red assay or XTT assay was performed as described earlier [104].

#### 3.3.4. Proliferation Assay

The impact of studied compounds on cell proliferation was estimated with DAPI staining. The 4 × 10^3^ cells per well were seeded into flat, clear bottom 96-well plates and stored in an incubator for 24 h at 37 °C to attach. After growth media removal, cells were treated with working solutions of studied dendrimer conjugates for 72 h at increasing concentrations. After plate centrifugation (5 min, 700× *g*), the medium was gently removed. The assay was performed as described [88].

#### 3.3.5. Cellular Accumulation of Labeled Conjugates

Dose-dependent cellular accumulation of studied dendrimer conjugates labeled with one molecule of fluorescein isothiocyanate (FITC) was performed using a microplate reader. All the studied cells (A549, U-118 MG, and HaCaT) were cultured as described in the proliferation protocol. Next, FITC-labeled conjugates were diluted in a complete culture medium with 10% FBS. Cells were treated in a range of increasing concentrations of dendrimers for 48 h. After centrifugation and washing with 1× PBS, cells were fixed in a 3,7% formalin solution, and nuclei were stained with 600 nM 33342 DAPI solution for 1 h. Then, plates were read with the Infinite M200 PRO Multimode Microplate Reader (TECAN Group Ltd., Männedorf, Switzerland) at 485/535 nm (FITC) and 360/460 nm (DAPI) against blank (wells with cells without conjugates). The fluorescence was expressed per the same number of cells by DAPI staining. Additionally, images from a fluorescence microscope (Delta Optical IB-100) were collected. 

#### 3.3.6. Toxicity to Caenorhabditis Elegans

The *Caenorhabditis elegans* nematode was used to estimate the in vivo activity of the synthesized dendrimer conjugates with paclitaxel, fulvestrant, and/or lapatinib. Nematodes wild-type culture (strain N2, variety Bristol) was maintained at 20 °C on NGM agar plates with *Escherichia coli* OP50 strain as a food source. C. elegans culture was synchronized by treatment with hypochlorite. Obtained eggs were left in M9 buffer at 21 °C to hatch until the next day. Then, L1 worms were placed on NGM plates with E. coli OP50 and left at 21 °C until reaching the L4 stage (approximately 44 h). L4 worms were transferred to 15 mL falcons by washing NGM plates twice with 5 mL water and centrifuged at 1500 rpm for 4 min. After supernatant aspiration, the pellet was re-suspended with 5 mL of complete S medium [105] followed by centrifugation. Subsequently, the density of nematode suspension was assessed according to Scanlan et al. [106]. Worms were suspended in complete S medium with E. coli OP50 (1:1000), 0.08% cholesterol (5 mg/mL in Et-OH), 1% penicillin–streptomycin, 1% nystatin, and 100 mM FUdR (at final concentration 200 µM) to obtain about 20 nematodes in 50 μL. FUdR was added to sterilize nematodes. 

After the transfer of nematodes to a 96-well plate (about 20 individuals in 50 μL), the solutions of the studied conjugates in a complete S medium were added (50 μL/well). The maximal DMSO final concentration was equal to 0.5% and had no significant influence on nematode viability. Five replicates were made for each concentration. The plate was incubated at 21 °C for seven days. During this time, live and dead worms were counted under the inverted microscope (Delta Optical IB-100). Microscopic images of some morphological changes were collected and analyzed in ImageJ 1.49v software to estimate nematodes’ body length.

#### 3.3.7. Statistical Analysis

For the cell culture assays, to estimate the differences between treated and non-treated control samples, a statistical analysis was performed using the nonparametric Kruskal–Wallis test. 

To analyze differences in nematode viability between the control, the non-treated group, and the nematodes incubated with conjugates, the Kaplan–Meier estimator was used. Statistically significant differences between the control and treated groups were determined with Gehan’s Wilcoxon test. *p* < 0.05 was considered statistically significant. All analyses and calculations were performed using Statistica 13.3 software (StatSoft, Tulsa, OK, USA).

## Figures and Tables

**Figure 1 molecules-28-06334-f001:**
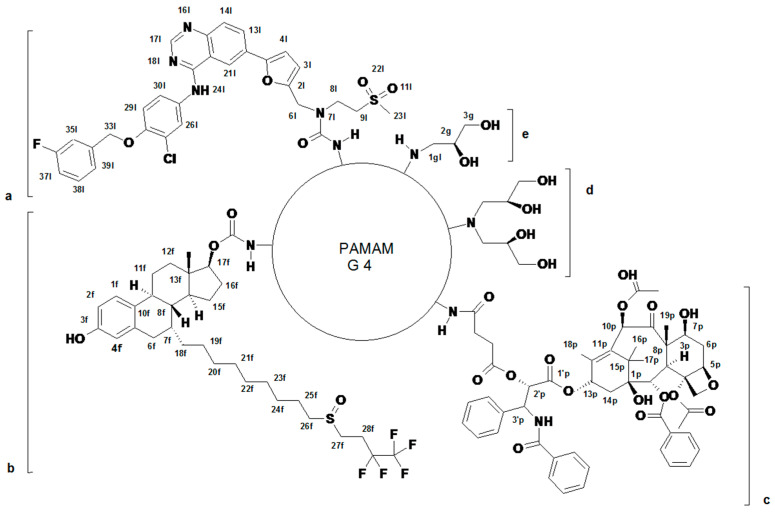
The schematic illustration of FITC single labeled PAMAM G4 conjugates with P (**c**), F (**b**), and L (**a**), covered with *R*-glycidol (**d**,**e**). The formula of fluorescein attached to G4 core is omitted for clarity. G4P: a, b = 0; c = 4; d = 50; e = 9; G4F: a = 0; b = 13; c = 0; d = 14; e = 36; G4L: a = 13; b, c = 0; d = 35; e = 15; G4PL: a = 10; b = 0; c = 4; d = 49; e = 0; G4PF: a = 0; b = 10; c = 4; d = 49; e = 0 G4PFL: a = 11; b = 11; c = 4; d = 37; e = 0. For details on synthetic pathways, see also Appendix A.

**Figure 2 molecules-28-06334-f002:**
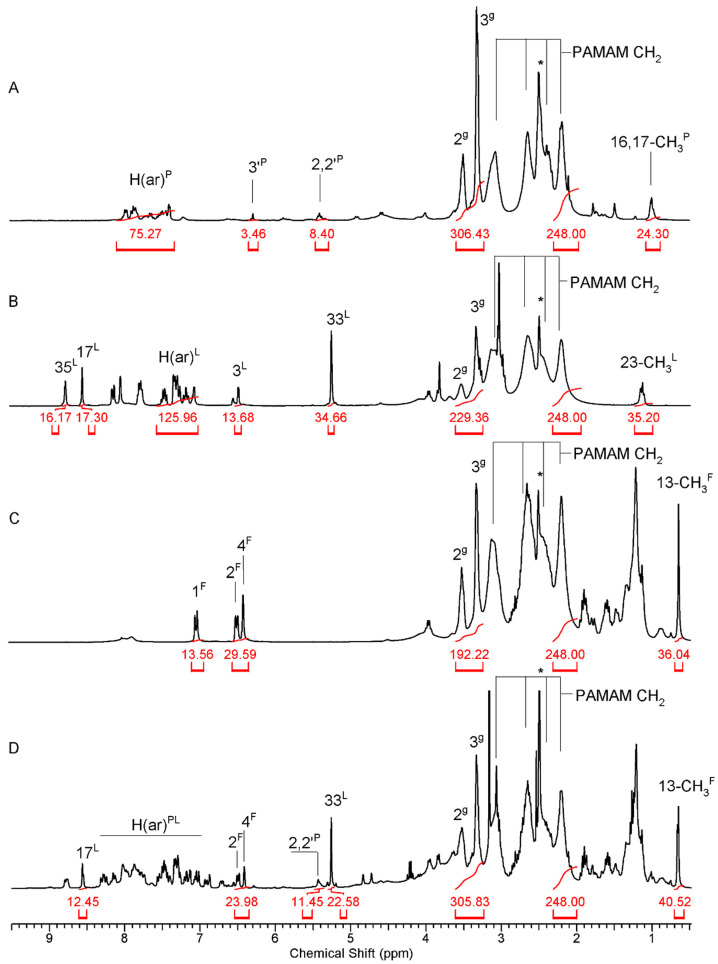
The ^1^H NMR spectra of the conjugates in DMSO-d_6_: (**A**)—G4*^4P109gl^, (**B**)—G4*^13L85gl^, (**C**)—G4*^13F64gl^, (**D**)—G4*^4P11F11P74gl^. For atom numbering see Figure 1. The residual CD_3_SOCD_2_H resonance is labeled with asterisk.

**Figure 3 molecules-28-06334-f003:**
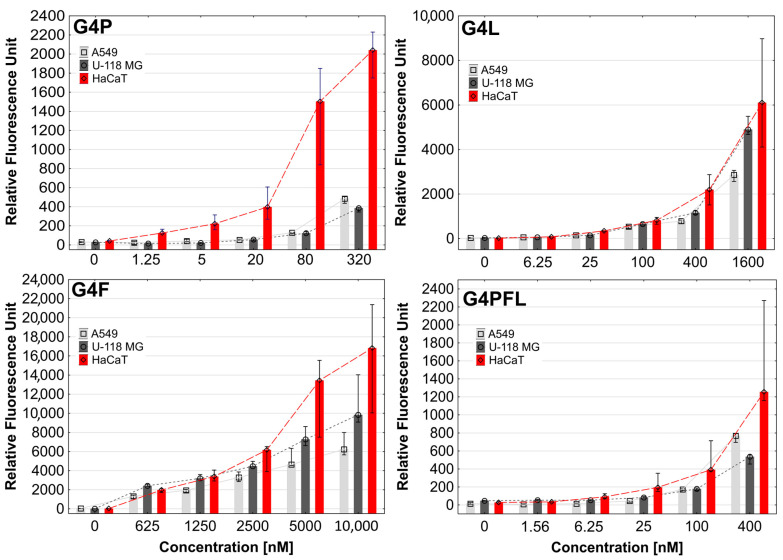
Cellular accumulation of G4P, G4F, G4L, and G4PFL FITC-labeled PAMAM dendrimer conjugates in A549, U-118 MG, and HaCaT cells after 48 h of incubation. Results were expressed as relative fluorescence units for the same number of cells. Bars indicate medians, whiskers first and third quartile.

**Figure 4 molecules-28-06334-f004:**
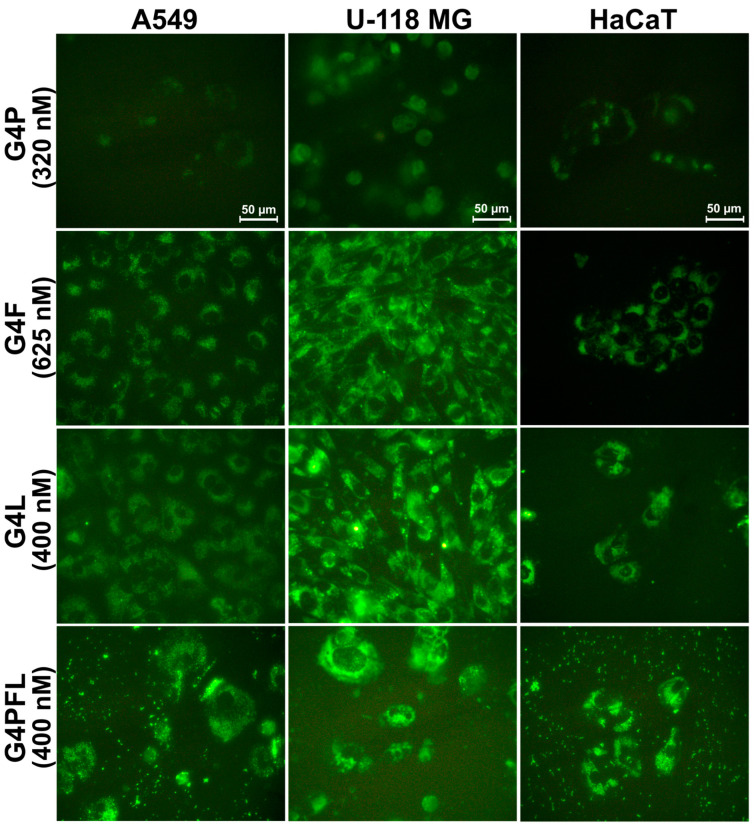
Images from fluorescence microscopy presenting cellular accumulation of fluorescently labeled G4P, G4F, G4L, and G4PFL conjugates after 48 h of incubation with similar values of nanomolar concentrations (described in parentheses). Green signal indicates FITC-labeled conjugates. Scale bar is equal to 50 µm.

**Figure 5 molecules-28-06334-f005:**
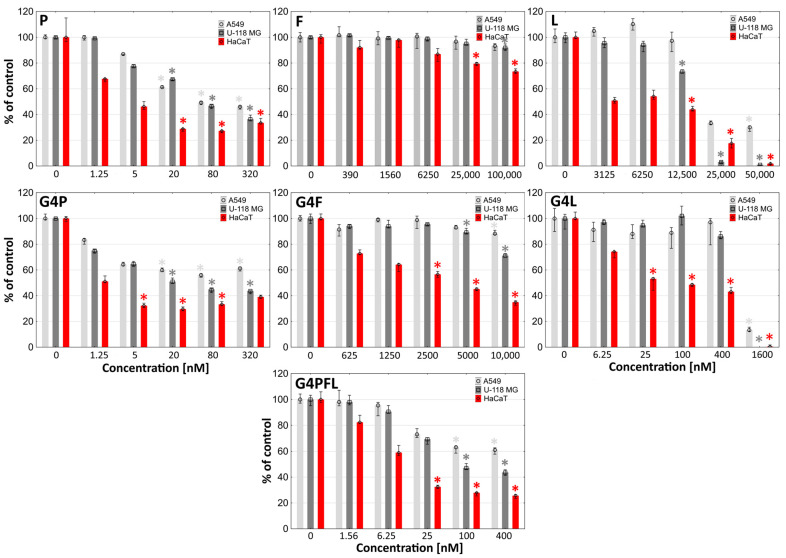
Cytotoxicity of paclitaxel (P), fulvestrant (F), lapatinib (L), and their conjugates with glycidylated PAMAM dendrimers G4P, G4F, G4L, and G4PFL against human cells: non-small-cell lung carcinoma (A549), glioma cells (U-118 MG), and immortalized keratinocytes (HaCaT) after 48 h incubation, estimated with an NR assay. Cell viability is expressed as median of a percent against non-treated control (control expressed as 100%). The whiskers are the lower (25%) and upper (75%) quartile ranges. * *p* ≤ 0.05; Kruskal–Wallis test (against non-treated control).

**Figure 6 molecules-28-06334-f006:**
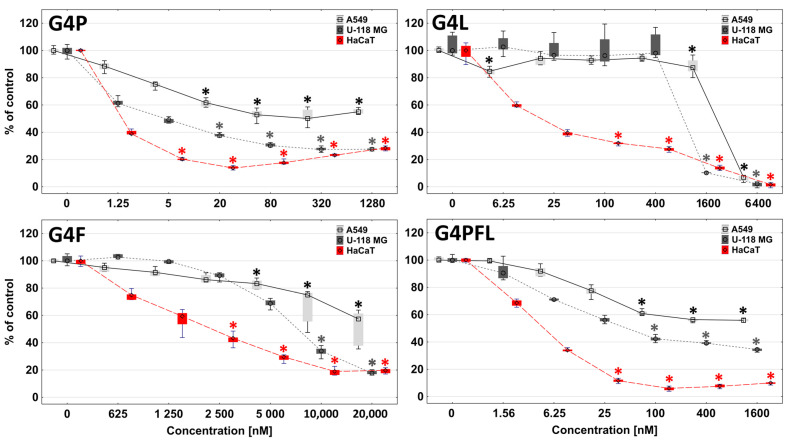
The influence of G4P, G4F, G4L, and G4PFL on A549, U-118 MG, and HaCaT on cell viability after 48 h treatment, estimated with an XTT assay. Cell viability is expressed as medians of a percent against non-treated control (control expressed as 100%). The boxes are the lower (25%) and upper (75%) quartile ranges, whiskers indicate minimum and maximum. * *p* ≤ 0.05; Kruskal–Wallis test (against non-treated control).

**Figure 7 molecules-28-06334-f007:**
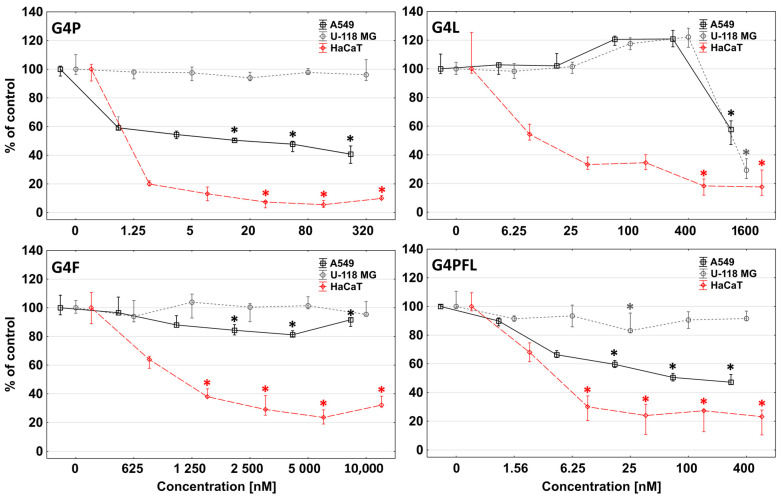
Anti-proliferative action of G4P, G4F, G4L, and G4PFL against A549, U-118 MG, and HaCaT cell lines after 72 h of incubation. Results are presented as medians (percentage of non-treated control). Whiskers indicate the lower (25%) and upper (75%) quartile ranges. * *p* ≤ 0.05, Kruskal–Wallis test (against non-treated control).

**Figure 8 molecules-28-06334-f008:**
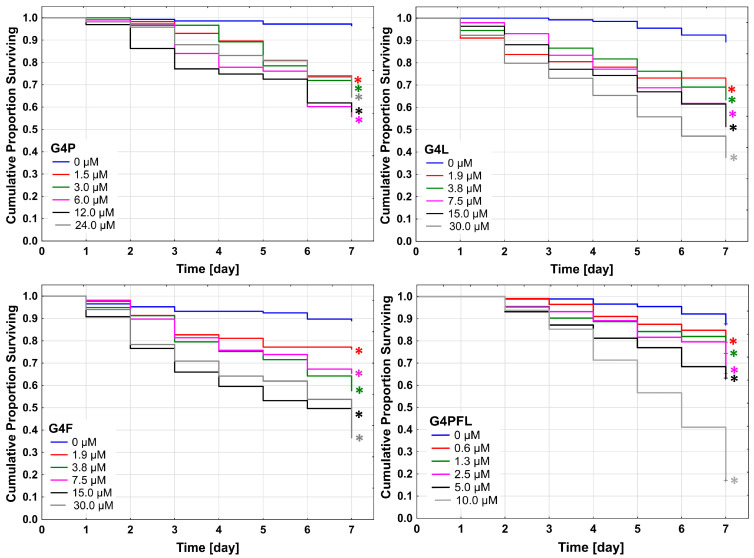
The Kaplan–Meier survival curves of *C. elegans* after 7 days of incubation with G4P, G4F, G4L, and G4PFL. Results are presented as cumulative proportion surviving. Statistically significant differences against DMSO-treated control obtained in Gehan’s Wilcoxon test are marked with asterisks * (*p* ≤ 0.05) in the colors corresponding to the tested concentrations.

**Figure 9 molecules-28-06334-f009:**
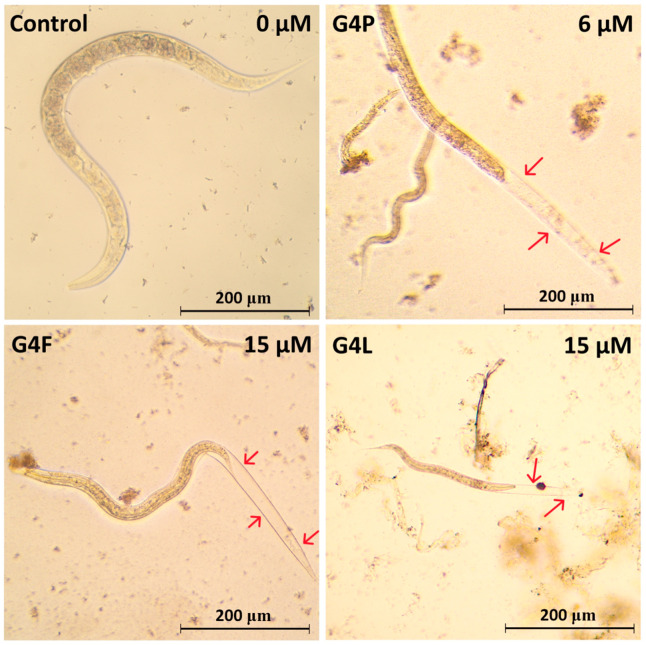
Morphology of *C. elegans* incubated for 7 days with glycidylated G4 PAMAM dendrimers. Red arrows show shrinked and molting individuals (cuticular molt). A decrease in body length of approximately 30% in the conjugate-treated groups in comparison to non-treated control is also visible.

**Table 1 molecules-28-06334-t001:** The composition and some properties of the conjugates of PAMAM G4 with lapatinib (L), fulvestrant (F), paclitaxel (P), and glycidol (gl). Solutions in dmso-d_6_. * means one equivalent of FITC attached through thiourea bond. Size and zeta potential were determined by DLS method. SD stands for standard deviation.

Conjugate	MW [kDa]	Size (SD) [nm]	Polydispersity Index PDI (SD)	ζ (SD) [mV]
G4*^126gl^ G4	19.1	5.12 (0.25)	0.080 (0.002)	4.35 (0.28)
G4*^4P109gl^ G4P	26.4	99.20 (7.28)	0.134 (0.016)	13.57 (0.43)
G4*^13F64gl^ G4F	27.6	138.23 (6.24)	0.155 (0.022)	38.02 (0.54)
G4*^13L85gl^ G4L	28.8	113.13 (4.64)	0.112 (0.014)	40.29 (0.74)
G4*^4P10L98gl^ G4PL	31.7	95.20 (4.46)	0.125 (0.010)	19.92 (0.53)
G4*^4P10F98gl^ G4PF	32.0	113.41 (5.48)	0.128 (0.015)	29.29 (0.72)
G4*^4P11F11L74gl^ G4PFL	37.5	105.47 (5.39)	0.159 (0.018)	34.33 (0.65)

**Table 2 molecules-28-06334-t002:** The half maximal inhibitory concentration (IC_50_) values determined following 48 h of treatment of A549, U-118 MG, or HaCaT cells with paclitaxel, fulvestrant, lapatinib, and their conjugates with glycidylated PAMAM G4 dendrimers G4P, G4F, G4L, and G4PFL: *p*—concentration of introduced P; *f*—concentration of introduced F; *l*—concentration of introduced L. The values of IC_50_ were calculated with AAT Bioquest IC_50_ calculator [79].

	IC_50_ [nM] NR Assay		
	A549	U-118 MG	HaCaT
Paclitaxel	50.35	55.56	2.53
Fulvestrant	≫100,00 *	≫100,000 *	≫100,000 *
Lapatinib	16,701.61	14,878.63	6150.48
G4P	≫320 * ≫1280 * *p*	25.28 101.12 *p*	1.63 6.52 *p*
G4F	≫1000 * ≫130,000 * *f*	≫10,000 * ≫130,000 * *f*	2927.15 38,052.97 *f*
G4L	1324.05 17,212.75 *l*	54,395 6527.4 *l*	57.87 765.31 *l*
G4PFL	≫400 * ≫400 * *p* ≫4400 * *fl*	78.70 314.80 *p* 865.70 *fl*	9.53 38.12 *p* 104.83 *fl*
	IC_50_ [nM] XTT Assay		
G4P	≫320 ≫1280 * *p*	3.14 12.56 *p*	0.99 3.96
G4F	>20,000 >260,000 *f*	6062.64 78,814.32 *f*	1282.15 16,667.95 *f*
G4L	3885.91 50,516.83 *l*	1147.86 14,922.18 *l*	76.90 999.70 *l*
G4PFL	>1600 >6400 *p* >17,600 *fl*	9.45 37.80 *p* 103.95 *fl*	2.00 8.00 *p* 22.00 *fl*

* cell viability was over 50% in tested range concentration. Lower IC50 values of appropriate conjugates are marked in red (comparison of NR and XTT assay).

**Table 3 molecules-28-06334-t003:** The half maximal lethal concentration (LC_50_) values determined following 7 days treatment of *Caenorhabditis elegans* with G4P, G4F, G4L, and G4PFL. Presented values were calculated with AAT Bioquest LC_50_ calculator [79].

Compound	LC_50_ [µM]
Mebendazole	4.00
G4P	>24.00
G4F	12.50
G4L	14.80
G4PFL	6.75

## Data Availability

Data available from the authors of the publication.

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
