# Peer review of "Exploring the Potential of Lapatinib, Fulvestrant, and Paclitaxel Conjugated with Glycidylated PAMAM G4 Dendrimers for Cancer and Parasite Treatment"

_molecules, 2023, doi:10.3390/molecules28176334_

Round 1

Author Response

ANSWERS to REVIEWER 1 and AMENDMENTS

REVIEWER 1

  1. In title: I suggest author to change the title into: “Exploring the Potential of Lapatinib, Fulvestrant, and Paclitaxel Conjugated with Glycidylated PAMAM G4 Dendrimers for Cancer and Parasite Treatment” or “Dendrimer-Based Drug Delivery for Glioma, Non-Small Cell Cancer, and Nematode Infections”

Thank you for your suggestion. We have changed the title of the paper into: “Exploring the Potential of Lapatinib, Fulvestrant, and Paclitaxel Conjugated with Glycidylated PAMAM G4 Dendrimers for Cancer and Parasite Treatment”.

  1. IC50 should be corrected to “IC50” (50 should subscripted) in entire manuscript.

Yes, we have done that.

  1. Please draw the synthetic schemes for conjugates and insert to the manuscript

Due to large size of synthetic chart we have added these three pathways of synthesis chart into Supplementary Material (Figure S1 A, B, and C) according to suggestion of Reviewer 3 (remark 2).

  1. Please make the conclusion in one paragraph.

We have done so in the revised version of the paper.

  1. No supporting materials were found.

Thank you for encouraging us to extend our submission. We always try to cut the text into small size, therefore additional data were not presented in the first version of the paper.

In revised version we have added the following Supplementary Figures:

Figure S1: Synthetic schemes for P, L, and F functionalization (activation) and covalent attachment into PAMAM G4 (mentioned in main text under Figure 1)

Figure S2: The heteronuclear 1H-13C HSQC and HMBC spectra of all conjugates are presented as combined maps, together with 1H and 13C spectra on 1-D traces, as well as the crucial one-bond scalar cross-peaks and multi-bond cross-peaks. We did that in order to unambiguous assign the 1H resonances of P, L, and F in conjugates. These resonances were then used to integrate against internal PAMAM G4 resonance to get the average stoichiometry of the conjugates.

Table S1: The details on DLS measurements performed in water and in phosphate buffer pH 5. Volume- and number-average diameters and zeta potential are specified for all conjugates, together with polydispersity index (PDI) and numerical detail results for d(V) for G4L and G4F in pH 5. These data are used in extended discussion in response to remarks of Reviewers 2 and 3.

Figure S3: Graphical illustration of d(V) and d(N) for conjugates, discussed in section 2.1 in the revised version

Figure S4: Graphical illustration of zeta potentials measured in pH 7 and 5, discussed in section 2.1 in the revised version.

Figure S5: Raw data on size distribution by intensity for conjugates: A – G4P; B – G4PFL; C- merged picture for all conjugates; D – the size distribution by volume for G4F depending on pH. These data were used to discussion in section 2.1 in the revised version.

Reviewer 2 Report

In this manuscript, the glycidylated PAMAM G4 dendrimer, substituted with fulvestrant, lapatinib and/or paclitaxel, were synthesized and characterized. Their antitumor and anthelmintic activity were evaluated in different cancer cells and Caenorhabditis elegans. The topic and finding of this manuscript is interesting and promising. However, there are still some issues need to be clarified before publishing.

1.     In the introduction part, the authors mentioned that the type of ER is important. However, it seems that the authors indicated in line 61-62 that ERα signaling has been found to be associated with 62 glioma progression, but ERβ with anti-cancer effects in glioblastoma. Whereas, in line 74-75, the authors mentioned that ERβ might impact also the progression and invasion of NSCLC. So it seems that different type of ER exhibited different function in different cancers. Am I right? If yes, why there is such kind of difference?

2.     Why the authors chose to use Caenorhabditis elegans as in vivo model to test the biological activity of constructed drug-dendrimer conjugates?

3.    In the synthesis part, whether the authors used other techniques to characterize dendrimer congjuates? Such as HPLC? Only using HNMR seems not so precise to confirm the structure of the target molecules.

4.     In Table 1, the particle size of G4F was much larger than that of other compounds. What might be the reason?

5.     In Table 1, we can find that the Zeta potential of the dendrimer drug conjugates were relatively positive. However according to the description and calculation in the synthesis part, the primary amine terminals of G4 PAMAM dendrimer have been substituted by drugs and fluorescent probe. So why the zeta potential of these dendirmer drug conjugates were positive? Whether the authors have tested the zeta potential of glycidylated G4 dendrimer without drug modification as control?

6.     In the cytotoxicity part (Figure 5), the synthesized PAMAM dendrimer drug conjugates exhibited significant influence in the proliferation of normal cells (HaCaT), potentially attributable to membrane receptor expression. Nevertheless, to ascertain the safety of the system, it is necessary to choose an appropriate normal cell for toxicity assessment.

7.     The association of these selected drugs does not appear to be strong, thereby necessitating further corroborating evidence to establish the necessity of their combined usage.

Author Response

ANSWERS to REVIEWER 3 and AMENDMENTS

REVIEWER 3

The manuscript molecules-2535313 "Potential usefulness of lapatinib, fulvestrant and paclitaxel conjugated with glycidylated PAMAM G4 dendrimers in glioma, non-small cell cancer and antinematode therapy" by Wołowiec and co-workers is describes the synthesis of four novel PAMAM G4 dendrimer based conjugates with lapatinib, fulvestrant and paclitaxel, and the study of their biological activity. The authors obtained interesting results, so I think this paper will be of interest to the readers of Molecules.

Some questions and comments:

  • The Introduction does not contain literature data on antinematode activity.

For paclitaxel, fulvestrant and lapatinib, no direct results of their toxicity to C. elegans are available. There are reports on the effect of paclitaxel on the structure of microtubules and embryos, and in the case of fulvestrant on the change in gene expression. Drugs from the lapatinib group (erlotinib, gefitinib) were tested for C. elegans vulval development. However, thank you for bringing to our attention the lack of information on C. elegans in the introduction. We have added relevant fragment for the introduction: “Potential anticancer therapeutics require also in vivo characterization of toxicity and mutagenicity. The use of vertebrate-based models should be preceded by tests on invertebrate animals. The nematode Caenorhabditis elegans is very attractive animal model to characterize the properties of anticancer therapeutics. The small size, ease of handling, and powerful genetic tools of C. elegans provide a sophisticated in vivo platform that combines the technical advantages of a microorganism with the greater biological complexity of a multicellular organism [55,56]. C. elegans has been used successfully to testing activity of EGFR tyrosine kinase inhibitors [57], selective estrogen receptor degraders (SERDs) [58] and microtubule-stabilizing drugs [59], therefore the biological activity of the synthesized constructs was studied also in vivo on C. elegans. Additionally, its usefulness as model for mixture toxicity testing was confirmed [60].

  • Please add schemes of step-by-step synthesis of target compounds for better understanding of the synthetic part of this work (can be in supplementary materials).

We have amended the submission by adding the initial steps of synthesis (drug functionalization or activation and addition to macromolecular carrier) in Supplementary Materials: Figures S1A, B, and C.

  • Lines 476-477. "The 1H, 13C NMR 1-D and 2-D spectra (1H-1H COSY and 1H-13C HSQC and HMBC) were recorded…". Unfortunately, I don't have access to the supplementary materials. The authors should add images of all NMR spectra of the novel synthesized compounds.

We have amended the submission by adding Figures S2A, S2B, S2C, and S2D illustrating the HSQC/HMBC heteronuclear correlation spectra 1H-13C. The spectra of glucoheptoamidated analogues were published previously:

Lewińska, A.; Wróbel, K.; Błoniarz, D.; Adamczyk-Grochala, J.; Wołowiec, S.; Wnuk, M. Lapatinib- and fulvestrant-PAMAM dendrimer conjugates promote apoptosis in chemotherapy-induced senescent breast cancer with different receptor status. Biomaterials Advances 2022, 140, 213047. https://doi.org/10.1016/j.bioadv.2022.213047

  • Since the synthesis was performed for the first time, please add a full interpretation of NMR spectra in the text of the experimental part. Some signals on 1H NMR spectra (Figure 2) are not correlated, although visually their intensity is higher than the marked signals. Therefore, the purity and individuality of the obtained compounds is in doubt. How were these characteristics confirmed?

Recently we have synthesized Lapatinib and Fulvestrant conjugates with PAMAM G3, which were further covered with glucoheptonoamide to erase remaining primary amine groups form the carrier: Lewińska, A.; Wróbel, K.; Błoniarz, D.; Adamczyk-Grochala, J.; Wołowiec, S.; Wnuk, M. Lapatinib- and fulvestrant-PAMAM dendrimer conjugates promote apoptosis in chemotherapy-induced senescent breast cancer with different receptor status. Biomaterials Advances 2022, 140, 213047. https://doi.org/10.1016/j.bioadv.2022.213047

In that paper we also made a detailed analysis of conjugated Lapatinib and Fulvestrant.

We amended this manuscript with 2-D HSQC and HMBC spectra of conjugates described here by adding the combined HSQC/HMBC maps with description of crucial peaks (Figures S2A, B, C, and D). This enabled us to assign the resonances in the 1-H NMR spectra of conjugates in order to choose the P, L, and F resonances for integration.

The truth is that observed stoichiometry is average. We used separated L, P, and F resonances to integrate against internal PAMAM G4 peak. Also the resonances of gl residues were separated enough to determine the number of gl residues attached to G4 by signal integration. The resonances used for integration are labeled in Figure 2. Some resonances from L, P, F have various intensity ratio than 1:2:3 due to dynamic behavior of these macromolecules; some of them are sharp if protons are far from “slowly” rotating arms of PAMAM G4 core, some are very broad. The latter were not considered for integration. Generally 1-H spectra are not bad for these objects. However, total assignment of 13-C resonances was not possible, because some cross-peaks in HSQC and especially in HMBC spectra unabled us total assignments. However, we did not see any additional 1-H resonances indicating any impurity, except residual proton resonance from dmso-d6 (labeled with asterisks) and water. In fact prior preparing sample of conjugate in dmso-d6 we dissolved conjugates in D2O and evaporated water in vacuo in order to quench ODH signal. That is the reason why amine proton resonances were also quenched and this resonance is visible only in Figure 2C.

We have no other methods to specific determination of the conjugate composition. Our attempts to obtain MALDI-TofF spectra gave no conclusive results because we observed only the fragmentation of these macromolecules. Neither we found any reliable MS results for PAMAM in literature, except PAMAM G1. Therefore we rely only on the 1-H NMR pattern and integral intensities of residues attached to unchanged PAMAM G4.

  • Please add DLS and zeta potential instrumental images in the supplementary materials. Is it distributional data by number, intensity, or volume? What about the polydispersity of the obtained supramolecular systems? What are the PDI values?

We have added details on size and zeta potential measurements for volume- and number-averaged size, and also zeta potential measured in pH 7 (water) and pH 5 (phosphate buffer) in Supplementary  Materials Figures S3, S4, S5 (raw data), and Table S1. The values of PDI were introduced into Table 1 (main text). The Discussion on size and zeta potential was amended in Section 2.1. (in red).

  • Lines 209-211. The authors compared the size of the aggregates of the synthesized conjugates with the size of a single PAMAM molecule and concluded incorrectly about the loading. Most likely the larger sizes were due to further self-assembly of "mono-molecules" into nanoassociates due to increased lipophilicity. The authors further made a similar assumption on lines 246-248. However, the results obtained require evidence. I recommend to investigate the obtained associates by microscopy (TEM or SEM).

We have changed confusing fragments considering the “loading” which is the word used rather to express number of molecules absorbed in nanocarrier. We modified Table 1 by introducing the results for the preglycidylated FITC-labeled PAMAM G4 in order to clearly show the differences in size of that molecule compare to the conjugates, which are at least one order of magnitude larger. We have also added the details on size and zeta determination in Supplementary Materials. We extended the Discussion related to size and zeta in (in red).

  • How correct is the comparison of the biological activity of the obtained conjugates with monomeric drugs, considering their high molecular masses (26.4-37.5 kDa) and the presence of several drug fragments in one macromolecule? If you recalculate taking this into account (e.g., concentration in µg/mL), you get completely different results.

Thank you for valuable remark. We have supplemented Table 2 by calculating the concentration of drugs introduced in appropriate conjugates. This allows for a clearer comparison of the activities of the conjugates and the drugs alone. Naturally, in the discussion we used quantities based on the equivalent concentrations in micromoles, so the final conclusions was changed:

In our study the binding of 13 residues of F to the glycidylated PAMAM G4 dendrimer resulted in an approximately 2-fold decrease of IC50 values for HaCaT dermal keratinocytes (comparing concentration of introduced drugs). Such an effect was not observed in the A549 cell line, where a decrease in cell viability under 50% was not no-ticed. The paclitaxel-containing conjugate did not induce a decrease in cell viability with increasing concentrations in HaCaT and A549 cells. This phenomenon was described as characteristic for paclitaxel [85].

In the case of lapatinib conjugate, its action was 2 and 8-folds stronger against gli-oma and keratinocyte cells, respectively, than equivalent of lapatinib alone (Fig. 5 and Tab. 2). Substitution of modified G4 PAMAM dendrimer vehicle with 4 residues of P resulted in about 2-fold lower attenuation of the conjugate effect compared to the drug alone for HaCaT and U-118 MG cell lines, and even 25-fold for A549 cells (Fig. 4). In the remaining cases, the effect of conjugates was proportional to the concentration of the introduced drugs”.

  • Table 3. Please add a well-known standard for comparison.

In table 3, we have placed the IC50 values (or more precisely LC50 - half maximal lethal concentration) for mebendazole - a standard antinematode drug and we added a comment in the discussion: “The G4PFL was the most toxic conjugate with 6.75 μM LC50 after seven days of incubation, and its toxicity was only slightly weaker than a common anthelmintic drug, mebendazole (LC50 = 4.0 μM).”

  • I recommend comparing the results obtained by the authors with previous results obtained by other scientific groups. Please add relevant paragraphs.

Relevant paragraph was added:

“Obtained results of toxicity are rather consistent with others. IC50 of paclitaxel for A549 and U-118 MG cells after 48 hours incubation was equal 11.0 and 21.1 nM, respectively [80,81]. IC50 of fulvestrant for A549 after 48 hours incubation was over 10 000 nM [78]. In case of lapatinib, IC50 was in range 5000 – 10000 nM for A549, (48 hours incubation), 8000 nM for U-118 MG and 200 nM for HaCaT (72 hours incubation) [82–84].”

  • Minor changes:

- Please add the zeta potential value to the abstract.

 We added zeta potential values into the abstract

- Line 468. "milimolar" should be "millimolar" (or mmole?).

Corrected

- Line 396. "Caenorhabditis elegan" should be in italic "Caenorhabditis elegan".

Caenorhabditis elegans is in italic in the revised version.

We would like to thank you the detailed reviewing. It helped us to improve the paper. We contacted our DLS coworker, discussed the results and eventually added her as coauthor 3 (dr Małgorzta Walczak) in the revised version.

Reviewer 3 Report

The manuscript molecules-2535313 "Potential usefulness of lapatinib, fulvestrant and paclitaxel conjugated with glycidylated PAMAM G4 dendrimers in glioma, non-small cell cancer and antinematode therapy" by Wołowiec and co-workers is describes the synthesis of four novel PAMAM G4 dendrimer based conjugates with lapatinib, fulvestrant and paclitaxel, and the study of their biological activity. The authors obtained interesting results, so I think this paper will be of interest to the readers of Molecules.

Some questions and comments:

1) The Introduction does not contain literature data on antinematode activity.

2) Please add schemes of step-by-step synthesis of target compounds for better understanding of the synthetic part of this work (can be in supplementary materials).

3) Lines 476-477. "The 1H, 13C NMR 1-D and 2-D spectra (1H-1H COSY and 1H-13C HSQC and HMBC) were recorded…". Unfortunately, I don't have access to the supplementary materials. The authors should add images of all NMR spectra of the novel synthesized compounds.

4) Since the synthesis was performed for the first time, please add a full interpretation of NMR spectra in the text of the experimental part. Some signals on 1H NMR spectra (Figure 2) are not correlated, although visually their intensity is higher than the marked signals. Therefore, the purity and individuality of the obtained compounds is in doubt. How were these characteristics confirmed?

5) Please add DLS and zeta potential instrumental images in the supplementary materials. Is it distributional data by number, intensity, or volume? What about the polydispersity of the obtained supramolecular systems? What are the PDI values?

6) Lines 209-211. The authors compared the size of the aggregates of the synthesized conjugates with the size of a single PAMAM molecule and concluded incorrectly about the loading. Most likely the larger sizes were due to further self-assembly of "mono-molecules" into nanoassociates due to increased lipophilicity. The authors further made a similar assumption on lines 246-248. However, the results obtained require evidence. I recommend to investigate the obtained associates by microscopy (TEM or SEM).

7) How correct is the comparison of the biological activity of the obtained conjugates with monomeric drugs, considering their high molecular masses (26.4-37.5 kDa) and the presence of several drug fragments in one macromolecule? If you recalculate taking this into account (e.g., concentration in µg/mL), you get completely different results.

8) Table 3. Please add a well-known standard for comparison.

9) I recommend comparing the results obtained by the authors with previous results obtained by other scientific groups. Please add relevant paragraphs.

10) Minor changes:

- Please add the zeta potential value to the abstract.

- Line 468. "milimolar" should be "millimolar" (or mmole?).

- Line 396. "Caenorhabditis elegan" should be in italic "Caenorhabditis elegan".

Please pay attention to the correct use of articles (sometimes it is missing). Please re-check English.

Author Response

(The authors gave the same response as above.)

Round 2

Reviewer 2 Report

The revised version has addressed our comments and questions. The current version can be published.

Author Response

Thank you very much for detailed reviewing our paper and final acceptation of the revised version

Reviewer 3 Report

The authors only partially corrected my comments. Unfortunately, I still don't have access to the supplementary materials. Therefore, I cannot evaluate synthetic schemes and images of any spectra. Therefore, I do not consider this manuscript ready for publication in Molecules.

Additional comments and recommendations:

1) Purity and individuality of the obtained compounds remain questionable. It is more likely that a mixture of a large number of partially substituted derivatives was obtained and used for biological experiments.

2) Microscopy images (TEM or SEM) should be added to confirm the formation of different sized aggregates.

3) Regarding my comment about the molecular mass of the studied objects. If practical use of the obtained results is supposed, then an increase in the total mass of the drug taken by an order of magnitude is not offset by a 2-fold improvement in their efficacy. Thus, I do not see that the modification studied by the authors has no positive effects in terms of biological activity.

Author Response

The authors only partially corrected my comments. Unfortunately, I still don't have access to the supplementary materials. Therefore, I cannot evaluate synthetic schemes and images of any spectra. Therefore, I do not consider this manuscript ready for publication in Molecules.

Additional comments and recommendations:

1) Purity and individuality of the obtained compounds remain questionable. It is more likely that a mixture of a large number of partially substituted derivatives was obtained and used for biological experiments.

2) Microscopy images (TEM or SEM) should be added to confirm the formation of different sized aggregates.

3) Regarding my comment about the molecular mass of the studied objects. If practical use of the obtained results is supposed, then an increase in the total mass of the drug taken by an order of magnitude is not offset by a 2-fold improvement in their efficacy. Thus, I do not see that the modification studied by the authors has no positive effects in terms of biological activity.

Ad 1: I am surprised that you have not seen the file Supplementary Materials which is separate pdf file deposited by us together with revised version of the manuscript. If necessary we can resubmit the schemes (Figures S1 A, S1B, and S1C) and 2-D spectra with full description (Figures S2A, S2B, S2C, S2D). Considering the “mixture” of species in post-reaction mixtures, we agree that number of P, L, or F molecules vary around the averaged formula determined from NMR signal integration. However, this dispersity is not large, as well as associate size determined by DLS. The data PDI (slightly larger than 0.1) included in Table 1 demonstrate it clearly. The detailed values of volume- and number-averaged size are given in additional materials in Figures S3 and S4. Furthermore some raw data on size and zeta potential are attached at the end of Supplementary Materials, which you did not see. Please load it and reconsider yours opinion. We also attached the Supplementary Materials at the end to this ANSWER.

We must also underline that this paper is about biology of obtained conjugates. The chemistry on those compounds is not described broadly, because we consider that full interpretation of NMR spectra is a regular chemical protocol. Description how we did it would take about 10 pages. The H-1 signal assignments was based on 2-D 1H-13-C correlations and comparison with the spectra of free Lapatinib, Paclitaxel and Fulvestrant in the same solvent. Spectral assignment is a regular chemical work at currently daily standard level. Please accept that presented 2-D spectra allowed us to define the positions and ultimately use integral intensity of drug residue resonances to determine the stoichiometry of the conjugates.

We appreciate your detailed verification of our paper.

Ad 2: We have not so immediate access to Electron Microscope. Previously we used another method to determine averaged size of this type of associates of G3-Nimesulide, namely Atomic Force Microscopy.

[Zaręba M, Sareło P, Kopaczyńska M, Białońska A, Uram Ł, Walczak M, Aebisher D, Wołowiec S. Mixed-Generation PAMAM G3-G0 Megamer as a Drug Delivery System for Nimesulide: Antitumor Activity of the Conjugates Against Human Squamous Carcinoma and Glioblastoma Cells. Int. J. Mol. Sci. 20 (2019) 4998.

doi:10.3390/ijms20204998]

There we saw various sizes of associates. Single deposited molecules of the G3-Nimesulide conjugates are visible in AFM as dots, the aggregates sized 8 – 80 nm. The aggregate size distribution depends on the way of sample preparation for imaging, which is different for AFM (deposition on mica) and different for SEM/TEM (deposition on carbon scaffold). Even though we have access to AFM the measurements, it would need to wait about 3 months to be performed. The AFM instrument belongs to another group, in another University, and it is used extensively for their purposes.

We do not see any advantage of imaging deposited associates of conjugate molecules. The most reliable way to follow size in solution or in suspension (this is the case for our conjugates with P, L, F) is DLS. Not surprisingly studied associates showed highly positive zeta potential, because it is additive value for monomolecular dispersion, though it is not proportional to the sum of individual zeta potentials. Furthermore, the value of zeta potential for associate depends on the number of molecules in associate (here the number of conjugate molecules is ca 1 000 – 10 000) considering the average diameter of associate (ca 100 nm) in comparison with single molecule (5 nm, entry 1 n Table 1).

Ad 3: We are not sure what are your expectations about doses of drug used by us to determine IC50. It is not the mass of PAMAM G4P, G4F, G4L or G4PFG conjugate which is relevant, rather the equivalent of introduced alone drugs, since we have indicated, that vehicle (glycidilated PAMAM dendrimer) was non-toxic up to 300000 nM concentration.

In the first version of the manuscript we calculated the IC50 for each of conjugates. According to your remark in previous review we recalculated it again for concentrations of total introduced drugs as if they were to be completely released from conjugates. Assuming that all molecules of L (or F or P) are released inside the targeted cell immediately after entering the cell, the recalculated IC50 for released drug is still lower than for “free” drug delivered without PAMAM G4 (see Table 2).

For instance:

  • the IC50 determined for Lapatinib alone on U-118 MG cells was 14878 nM, while that calculated for G4L conjugate it was 543.95 nM. In G4L total number of introduced residues of Lapatinib was equal 13, therefore we recalculated IC50 which was 13-fold higher (13×95 nM =  6527 nM). This means two-fold increase of efficacy (14878/6527= 2.28) of conjugate-delivered drug related to “free” drug.

Therefore, we do not understand, why you did not accept our approach. The same way of reasoning was appreciated by the reviewers and readers of our previous work: [Ł. Uram, A. Filipowicz, M. Misiorek, N. Pieńkowska, J. Markowicz, E. Wałajtys-Rode, S. Wołowiec. Biotinylated PAMAM G3 dendrimer conjugated with celecoxib and/or Fmoc-l-Leucine and its cytotoxicity for normal and cancer human cell lines, European Journal of Pharmaceutical Sciences, Volume 124, 2018, Pages 1-9, https://doi.org/10.1016/j.ejps.2018.08.019.

Please accept our approach.

Thank you for your criticism and time spent with our manuscripts.

Please fing also the attached Answer and Supplementary Materials

Round 3

Reviewer 3 Report

I thank the authors for answering my questions and improving the manuscript